# On Conformal and Concircular Diffeomorphisms of Eisenhart's Generalized Riemannian Spaces

**Miloš Z. Petrović** [1] , **Mića S. Stanković** [2,*] **and Patrik Peška** [3]

[1] Department of Mathematics and Computer Science, Faculty of Agriculture in Kruševac, University of Niš, Kosančićeva 4, 37000 Kruševac, Serbia

[2] Department of Mathematics, Faculty of Sciences and Mathematics, Višegradska 33, 18000 Niš, Serbia

[3] Department of Algebra and Geometry, Faculty of Science, Palacký University, 17. listopadu 1192/12, 771 46 Olomouc, Czech Republic

* Correspondence: stmica@ptt.rs

**Abstract:** We consider conformal and concircular mappings of Eisenhart's generalized Riemannian spaces. We prove conformal and concircular invariance of some tensors in Eisenhart's generalized Riemannian spaces. We give new generalizations of symmetric spaces via Eisenhart's generalized Riemannian spaces. Finally, we describe some properties of covariant derivatives of tensors analogous to Yano's tensor of concircular curvature in Eisenhart symmetric spaces of various kinds.

**Keywords:** conformal mapping; Eisenhart's generalized Riemannian space; Weyl conformal tensor; Yano's concircular curvature; symmetric space

## 1. Introduction

A non-symmetric basic tensor was used by several authors as the main axiom of the theory which is nowdays called a non-symmetric gravitational theory [1]. Reference [2] formally introduced a generalized Riemannian space as a differentiable manifold endowed with a non-symmetric basic tensor. The non-symmetric linear connection on a generalized Riemannian space $(M, g)$ is explicitly determined by the compatibility condition with the symmetric part $\underline{g}$ of non-symmetric metric $g$. References [3,4] found new curvature tensors of a non-symmetric linear connection.

A mapping between two generalized Riemannian spaces is said to be a conformal mapping if it preserves angles between curves of these spaces. Some physical characteristics of conformal mappings were given in [5]. Geodesic mappings and their generalizations is an active research field, see for instance [6–15]. Some conformal and projective invariants of Riemannnian manifolds were obtained by Reference [16,17]. Recently, some very interesting remarks on the converse of Weyl's conformal theorem were given by [18]. On the other hand [19] had proved conformal invariace of dual projective curvature tensor.

Let us recall that a geodesic circle on a differentiable manifold is a curve which has constant first curvature and vanishing second curvature. A conformal mapping $f : M \to \overline{M}$ which preserves the geodesic circles of the manifold $M$ is said to be a concircular mapping [20].

Eisenhart's non-symmetric metric is the fundamental metric tensor in the non-symmetric gravitational theory (NGT). Recently, some problems and hopes related with the non-symmetric gravity were given by [21]. In the papers [22,23] and the papers that follow these ones, the authors studied conformal and concircular mappings of generalized Riemannian spaces with assumption that these mappings were preserving the torsion tensor. In the present paper we studied conformal and concircular mappings of generalized Riemannian spaces without any of the restrictive assumptions and find some tensors that are invariant with respect to these mappings.

In the existing literature there exist various generalizations of symmetric spaces. We define some new kinds of symmetric spaces with torsion by taking into account five curvature tensors of Eisenhart's generalized Riemannian spaces and four kinds of covariant derivative.

## 2. Conformal Mappings of Generalized Riemannian Spaces

A generalized Riemannian space in Eisenhart's sense $(M, g)$ is a differentiable manifold $M$ endowed with a non-symmetric metric $g$ which can be described thorough its symmetric and skew-symmetric parts as [2]

$$g(x, y) = \underline{g}(x, y) + \underset{\vee}{g}(x, y),$$

where $\underline{g}$ denotes the symmetric part of $g$ and $\underset{\vee}{g}$ denotes the skew-symmetric part of $g$, i.e.,

$$\underline{g}(x, y) = \frac{1}{2}\left(g(x, y) + g(y, x)\right) \quad \text{and} \quad \underset{\vee}{g}(x, y) = \frac{1}{2}\left(g(x, y) - g(y, x)\right).$$

On a generalized Riemannian space in Eisenhart's sense a non-symmetric linear connection $\underset{1}{\nabla}$ which is compatible with the symmetric part $\underline{g}$ of generalized Riemannian metric $g$ is explicitly determined by [2]

$$2\underline{g}(\underset{1}{\nabla}_x y, z) = x g(y, z) + y g(z, x) - z g(y, x).$$

The non-symmetric linear connection $\underset{1}{\nabla}$ can be described thorough its symmetric part $\underset{0}{\nabla}$ and the torsion tensor $T$ as

$$\underset{1}{\nabla}_x y = \underset{0}{\nabla}_x y + \frac{1}{2} T(x, y), \tag{1}$$

where the symmetric part $\nabla$ and the torsion tensor $T$ of non-symmetric linear connection $\underset{1}{\nabla}$ are respectively determined by

$$\underset{0}{\nabla}_x y = \frac{1}{2}\left(\underset{1}{\nabla}_x y + \underset{1}{\nabla}_y x\right) \text{ and } T(x, y) = \underset{1}{\nabla}_x y - \underset{1}{\nabla}_y x.$$

A generalized Riemannian space is endowed with a non-symmetric linear connection, so there exist four kinds of covariant differentiation [24]:

$$\underset{1}{\nabla}_j a_i^h = \partial_j a_i^h + \Gamma_{pj}^h a_i^p - \Gamma_{ij}^p a_p^h, \qquad \underset{3}{\nabla}_j a_i^h = \partial_j a_i^h + \Gamma_{pj}^h a_i^p - \Gamma_{ji}^p a_p^h,$$

$$\underset{2}{\nabla}_j a_i^h = \partial_j a_i^h + \Gamma_{jp}^h a_i^p - \Gamma_{ji}^p a_p^h, \qquad \underset{4}{\nabla}_j a_i^h = \partial_j a_i^h + \Gamma_{jp}^h a_i^p - \Gamma_{ij}^p a_p^h.$$

Let $(M, g)$ and $(\overline{M}, \overline{g})$ be two generalized Riemannian spaces of dimension $n > 2$. We can consider the manifolds $M$ and $\overline{M}$ in the common coordinate system with respect to the mapping $f : M \to \overline{M}$. In what follows we will assume that all mappings under consideration are diffeomorphisms, which particularly mean that these mappings are bijections. In this coordinate system the corresponding points $p \in M$ and $f(p) \in \overline{M}$ have the same coordinates and we can consider the connection deformation tensor

$$P(x, y) = \underset{0}{\overline{\nabla}}_x y - \underset{0}{\nabla}_x y,$$

where $\underset{0}{\nabla}$ and $\underset{0}{\overline{\nabla}}$ are the Levi–Civita connections of metrics $\underline{g}$ and $\overline{\underline{g}}$ that are symmetric parts of generalized Riemannian metrics $g$ and $\overline{g}$, respectively.

The symmetric part $\underline{g}$ of a generalized Riemmanian metric $g$ is of a non-degenerate symmetric bilinear form and defines an inner product on the tangent space of a generalized Riemannian space. From the definition of a conformal mapping, one can find that (see page 237 in [11])

$$\frac{\underline{g}(x,y)}{\sqrt{\underline{g}(x,x)\underline{g}(y,y)}} = \frac{\overline{\underline{g}}(x,y)}{\sqrt{\overline{\underline{g}}(x,x)\overline{\underline{g}}(y,y)}}, \tag{2}$$

where $x$ and $y$ are tangent vectors of two intersecting curves in the intersection point.

From (2) it follows that [11]

$$\overline{\underline{g}} = e^{2\sigma}\underline{g}, \tag{3}$$

where $\sigma$ is a function on $M$.

In [22] the authors made the assumption

$$\overline{g} = e^{2\sigma}g, \tag{4}$$

which further implies (3) and the same equation is valid for the skew-symmetric parts (which are not metrics) of the metrics $g$ and $\overline{g}$, i.e.,

$$\overset{\vee}{\overline{g}} = e^{2\sigma}\overset{\vee}{g}. \tag{5}$$

Obviously, the condition (3) is weaker than the condition (4) and it is independent of condition (5).

If there exists a conformal mapping $f : M \to \overline{M}$ between generalized Riemannian spaces $(M, g)$ and $(\overline{M}, \overline{g})$, then the connection deformation tensor $P(x, y)$ takes form [11]

$$P(x,y) = \mathrm{d}(\sigma)(x)y + \mathrm{d}(\sigma)(y)x - \underline{g}(x,y)\underset{0}{\nabla}\sigma, \tag{6}$$

where $\mathrm{d}(\sigma)$ is a gradient vector field.

S. M. Minčić cosidered four kinds of covariant derivatives $\underset{\theta}{\nabla}, \theta = 1, \ldots, 4$ and examined various Ricci type identities [24]. Also, he showed that among the twelve curvature tensors which appeared in the Ricci type identities, there exist five which are linearly independent [4]:

$$
\begin{aligned}
\underset{1}{R}^h{}_{ijk} &= L^h_{ij,k} - L^h_{ik,j} + L^p_{ij}L^h_{pk} - L^p_{ik}L^h_{pj}; \\
\underset{2}{R}^h{}_{ijk} &= L^h_{ji,k} - L^h_{ki,j} + L^p_{ji}L^h_{kp} - L^p_{ki}L^h_{jp}; \\
\underset{3}{R}^h{}_{ijk} &= L^h_{ij,k} - L^h_{ki,j} + L^p_{ij}L^h_{kp} - L^p_{ki}L^h_{pj} + L^p_{kj}(L^h_{pi} - L^h_{ip}); \\
\underset{4}{R}^h{}_{ijk} &= L^h_{ij,k} - L^h_{ki,j} + L^p_{ij}L^h_{kp} - L^p_{ki}L^h_{pj} + L^p_{jk}(L^h_{pi} - L^h_{ip}); \\
\underset{5}{R}^h{}_{ijk} &= \frac{1}{2}\big(L^h_{ij,k} - L^h_{ik,j} + L^h_{ji,k} - L^h_{ki,j} + L^p_{ij}L^h_{pk} + L^p_{ji}L^h_{kp} \\
&\quad - L^p_{ik}L^h_{jp} - L^p_{ki}L^h_{pj}\big).
\end{aligned}
\tag{7}
$$

Corresponding Ricci tensors are defined by

$$\underset{\theta}{Ric}_{ij} = \underset{\theta}{R}^p{}_{ijp}, \tag{8}$$

where $\theta = 1, \ldots, 5$.

Reference [3] gave the geometric meaning of the curvature tensors $\underset{\theta}{R}^h{}_{ijk}, \theta = , \ldots, 4$ by taking into account parallel displacement with respect to covariant derivatives $\underset{\theta}{\nabla}, \theta = 1, 2$. The geometric meaning of the fifth curvature tensor $\underset{5}{R}^h{}_{ijk}$ was given by S. M. Minčić.

**Definition 1.** *The scalar curvature $\underset{\theta}{\tau}$ of kind $\theta \in \{1, \ldots, 5\}$ of a generalized Riemannian space $(M, g)$ is defined by*

$$\underset{\theta}{\tau} = g^{ij}\underset{\theta}{Ric}_{ij},$$

*where $\underset{\theta}{Ric}_{ij} = \underset{\theta}{R}^{p}_{ijp}$ is the Ricci tensor of kind $\theta \in \{1, \ldots, 5\}$ given by* (8).

In the same manner as it was done for the Riemannian curvature tensors in [11] we can find the relation between the components $\underset{\theta}{R}^{h}_{ijk}$ and $\underset{\theta}{\overline{R}}^{h}_{ijk}$ of curvature tensors $\underset{\theta}{R}(x,y)z$ and $\underset{\theta}{\overline{R}}(x,y)z$, $\theta \in \{1, \ldots, 5\}$, given by (7) with respect to the conformal mapping $f : M \to \overline{M}$ between generalized Riemannian spaces $(M, g)$ and $(\overline{M}, \overline{g})$ [11]:

$$\underset{\theta}{\overline{R}}^{h}_{ijk} = \underset{\theta}{R}^{h}_{ijk} + \delta^{h}_{k}\sigma_{ij} - \delta^{h}_{j}\sigma_{ik} + \sigma^{h}_{k}g_{\underline{ij}} - \sigma^{h}_{j}g_{\underline{ik}} + \left(\delta^{h}_{k}g_{\underline{ij}} - \delta^{h}_{j}g_{\underline{ik}}\right)\Delta_{1}\sigma \tag{9}$$

where

$$\sigma_{ij} = \underset{0}{\nabla}_{j}\sigma_{i} - \sigma_{i}\sigma_{j}, \quad \sigma^{h}_{k} = g^{hp}\sigma_{pk}, \quad \Delta_{1}\sigma = g^{pq}\sigma_{p}\sigma_{q}. \tag{10}$$

Contracting (9) on the indices $h$ and $k$ we obtain that [11]

$$\underset{\theta}{\overline{Ric}}_{ij} = \underset{\theta}{Ric}_{ij} - (n-2)\sigma_{ij} - \left(\Delta_{2}\sigma + (n-2)\Delta_{1}\sigma\right)g_{\underline{ij}}, \tag{11}$$

where $\underset{\theta}{\overline{Ric}}_{ij} = \underset{\theta}{\overline{R}}^{p}_{ijp}$ and $\underset{\theta}{Ric}_{ij} = \underset{\theta}{R}^{p}_{ijp}$ are the Ricci tensors and $\Delta_{2}\sigma = g^{pq}\nabla_{q}\sigma_{p}$.

By contracting (11) with $g^{ij}$ we get [11]

$$e^{2\sigma}\underset{\theta}{\overline{\tau}} = \underset{\theta}{\tau} - 2(n-1)\Delta_{2}\sigma - (n-1)(n-2)\Delta_{1}\sigma, \tag{12}$$

where $\underset{\theta}{\tau} = g^{ij}\underset{\theta}{Ric}_{ij}$ and $\underset{\theta}{\overline{\tau}} = \overline{g}^{ij}\underset{\theta}{\overline{Ric}}_{ij}$ are the scalar curvatures of metrics $g$ and $\overline{g}$, respectively.

From (9), (11) and (12) it follows that the tensors $\underset{\theta}{C}(x,y)z$, $\theta = 1, \ldots, 5$ given below are invariant with respect to the conformal mapping $f : M \to \overline{M}$ between generalized Riemannian spaces $(M, g)$ and $(\overline{M}, \overline{g})$:

$$\underset{\theta}{C}(x,y)z = \underset{\theta}{R}(x,y)z + \underset{\theta}{L}(x,y)z, \quad \theta \in \{1, \ldots, 5\}, \tag{13}$$

where the components $\underset{\theta}{L}^{h}_{ijk}$ of tensors $\underset{\theta}{L}(x,y)z$, $\theta \in \{1, \ldots, 5\}$ are determined by

$$\underset{\theta}{L}^{h}_{ijk} = \frac{1}{n-2}\left(\delta^{h}_{j}\left(\underset{\theta}{Ric}_{ik} - \frac{\underset{\theta}{\tau}}{2(n-1)}g_{\underline{ik}}\right) - \delta^{h}_{k}\left(\underset{\theta}{Ric}_{ij} - \frac{\underset{\theta}{\tau}}{2(n-1)}g_{\underline{ij}}\right)\right.$$
$$\left. - g^{hp}\left(\underset{\theta}{Ric}_{pk} - \frac{\underset{\theta}{\tau}}{2(n-1)}g_{\underline{pk}}\right)g_{\underline{ij}} + g^{hp}\left(\underset{\theta}{Ric}_{pj} - \frac{\underset{\theta}{\tau}}{2(n-1)}g_{\underline{pj}}\right)g_{\underline{ik}}\right).$$

**Theorem 1.** *Let $f : M \to \overline{M}$ be a conformal mapping between two generalized Riemannian spaces $(M, g)$ and $(\overline{M}, \overline{g})$ of dimension $n > 2$. Then the tensors $\underset{\theta}{C}(x,y)z$, $\theta = 1, \ldots, 5$ defined by (13) are invariant with respect to the mapping $f$.*

The tensors $\underset{\theta}{C}(x,y)z$, $\theta = 1, \ldots, 5$ are analogous to the Weyl conformal curvature tensor. In Theorems 2 and 3 we will prove that the tensors $\underset{\theta}{C}(x,y)z$, $\theta = 1, \ldots, 5$ given by (13) have the same properties as the curvature tensors $\underset{\theta}{R}(x,y)z$, $\theta = 1, \ldots, 5$.

**Theorem 2.** *Let $(M, g)$ and $(\overline{M}, \overline{g})$ be two generalized Riemannian spaces of dimension $n > 2$ and $f : M \to \overline{M}$ be a conformal mapping. Then the tensors $\underset{\theta}{C}(x, y)z$, $\theta = 1, \dots, 5$ determined by* (13) *satisfy:*

$$\text{(i)} \qquad \underset{1}{C}(x, y)z + \underset{1}{C}(y, x)z = 0,$$

$$\text{(ii)} \qquad \underset{2}{C}(x, y)z + \underset{2}{C}(y, x)z = 0,$$

$$\text{(iii)} \qquad \underset{3}{C}(x, y)z + \underset{3}{C}(y, x)z = (\underset{0}{\nabla}_x T)(z, y) + (\underset{0}{\nabla}_y T)(z, x),$$

$$\text{(iv)} \qquad \underset{4}{C}(x, y)z + \underset{4}{C}(y, x)z = (\underset{0}{\nabla}_x T)(z, y) + (\underset{0}{\nabla}_y T)(z, x),$$

$$\text{(v)} \qquad \underset{5}{C}(x, y)z + \underset{5}{C}(y, x)z = 0.$$

**Proof.** First, let us give the proof for the case $\theta = 1$. In this case formula (13) reads

$$\underset{1}{C}(x, y)z = \underset{1}{R}(x, y)z + \underset{1}{L}(x, y)z,$$

which further implies

$$\underset{1}{C}(x, y)z + \underset{1}{C}(y, x)z = \underset{1}{R}(x, y)z + \underset{1}{R}(y, x)z + \underset{1}{L}(x, y)z + \underset{1}{L}(y, x)z = 0,$$

where we used the following properties of the curvature tensor $\underset{1}{R}(x, y)z$ and the tensor $\underset{1}{L}(x, y)z$ [25]

$$\underset{1}{R}(x, y)z + \underset{1}{R}(y, x)z = 0 \quad \text{and} \quad \underset{1}{L}(x, y)z + \underset{1}{L}(y, x)z = 0.$$

This completes the proof of part $(i)$. Part $(ii)$ can be proved analogously. To prove part $(iii)$ let us observe that the curvature tensor $\underset{3}{R}(x, y)z$ does not have the same properties as the curvature tensors $\underset{1}{R}(x, y)z$ and $\underset{2}{R}(x, y)z$, i.e., [25]

$$\underset{3}{R}(x, y)z \neq -\underset{3}{R}(y, x)z,$$

or more precisely

$$\underset{3}{R}(x, y)z + \underset{3}{R}(y, x)z = (\underset{0}{\nabla}_x T)(z, y) + (\underset{0}{\nabla}_y T)(z, x).$$

Therefore,

$$\underset{3}{C}(x, y)z + \underset{3}{C}(y, x)z = \underset{3}{R}(x, y)z + \underset{3}{R}(y, x)z + \underset{3}{L}(x, y)z + \underset{3}{L}(y, x)z$$
$$= (\underset{0}{\nabla}_x T)(z, y) + (\underset{0}{\nabla}_y T)(z, x),$$

which completes the proof of part $(iii)$. Parts $(iv)$ and $(v)$ can be proved in the same manner. $\square$

Let us denote

$$\underset{x, y, z}{\mathfrak{S}} A(x, y, z) = A(x, y, z) + A(y, z, x) + A(z, x, y),$$

where $A$ is an arbitrary $(1, 3)$ tensor field.

**Theorem 3.** *Let $(M, g)$ and $(\overline{M}, \overline{g})$ be two generalized Riemannian spaces of dimension $n > 2$ and $f : M \to \overline{M}$ be a conformal mapping. Then the tensors $\underset{\theta}{C}(x, y)z$, $\theta = 1, \dots, 5$ determined by* (13) *satisfy*

$$\underset{x, y, z}{\mathfrak{S}} \underset{1}{C}(x, y)z = \underset{x, y, z}{\mathfrak{S}} \underset{1}{L}(x, y)z + (\underset{0}{\nabla}_x T)(z, y) + (\underset{0}{\nabla}_z T)(y, x) + (\underset{0}{\nabla}_y T)(x, z)$$

$$+ \frac{1}{2} \Big( T(T(z, y), x) + T(T(x, z), y)$$

$$+ T(T(y, x), z) \Big);$$

$$\underset{x,y,z\,2}{\mathfrak{S}}\,C(x,y)z = \underset{x,y,z\,2}{\mathfrak{S}}\,L(x,y)z - (\underset{0}{\nabla}_x T)(z,y) - (\underset{0}{\nabla}_z T)(y,x) - (\underset{0}{\nabla}_y T)(x,z)$$
$$+\frac{1}{2}\Big(T(T(z,y),x) + T(T(x,z),y)$$
$$+ T(T(y,x),z)\Big);$$

$$\underset{x,y,z\,3}{\mathfrak{S}}\,C(x,y)z = \underset{x,y,z\,3}{\mathfrak{S}}\,L(x,y)z + \frac{1}{2}\Big(T(T(z,y),x) + T(T(x,z),y)$$
$$+ T(T(y,x),z)\Big)$$
$$-\frac{1}{2}\Big(T(T(y,x),z) + T(T(x,z),y)$$
$$+ T(T(z,y),x)\Big);$$

$$\underset{x,y,z\,4}{\mathfrak{S}}\,C(x,y)z = \underset{x,y,z\,4}{\mathfrak{S}}\,L(x,y)z - \frac{1}{2}\Big(T(T(z,y),x) + T(T(x,z),y)$$
$$+ T(T(y,x),z)\Big)$$
$$+\frac{1}{2}\Big(T(T(y,x),z) + T(T(x,z),y)$$
$$+ T(T(z,y),x)\Big);$$

$$\underset{x,y,z\,5}{\mathfrak{S}}\,C(x,y)z = \underset{x,y,z\,5}{\mathfrak{S}}\,L(x,y)z - \frac{1}{2}\Big(T(T(z,y),x) + T(T(x,z),y) + T(T(y,x),z)\Big).$$

*Concircular Mappings of Generalized Riemannian Spaces*

We shall consider concircular mappings between generalized Riemannian spaces in Eisenhart's sense. In what follows we shall prove that there exist some tensors invariant with respect to these mappings. Let $(M,g)$ and $(\overline{M},\overline{g})$ be two generalized Riemannian spaces of dimension $n > 2$. If a conformal mapping $f : M \to \overline{M}$ is concircular then the tensor $\sigma_{ij}$ determined in Equation (10) satisfies [11]

$$\sigma_{ij} = \omega g_{\underline{ij}},$$

where $\omega$ is a function.

**Theorem 4.** *Let $(M,g)$ and $(\overline{M},\overline{g})$ be two n-dimensional $(n > 2)$ generalized Riemannian spaces and $f : M \to \overline{M}$ be a concircular mapping. Then the tensors $\underset{\theta}{Y}(x,y)z$ and $\underset{\theta}{E}(x,y)$, $\theta \in \{1,\dots,5\}$ given below are invariant with respect to the mapping $f$*

$$\underset{\theta}{Y}(x,y)z = \underset{\theta}{R}(x,y)z - \frac{\underset{\theta}{\tau}}{n(n-1)}\big(\underline{g}(x,z)y - \underline{g}(y,z)x\big), \quad \theta\{1,\dots,5\}, \tag{14}$$

$$\underset{\theta}{E}(x,y) = \underset{\theta}{Ric}(x,y) - \frac{1}{n}\underset{\theta}{\tau}\underline{g}(x,y), \quad \theta\{1,\dots,5\}, \tag{15}$$

*where $x,y,z \in T_p(M)$.*

**Proof.** Since the mapping $f : M \to \overline{M}$ between two generalized Riemannian spaces $(M,g)$ and $(\overline{M},\overline{g})$ is concircular the formulas (9), (11) and (12) respectively become

$$\underset{\theta}{\overline{R}}(x,y)z = \underset{\theta}{R}(x,y)z - \omega\big(\underline{g}(y,z)x - \underline{g}(x,z)y\big), \quad \theta\{1,\dots,5\},$$

$$\overline{\underset{\theta}{Ric}}(x,y) = \underset{\theta}{Ric}(x,y) + (n-1)\omega \underline{g}(x,y), \quad \theta\{1,\ldots,5\},$$

$$e^{2\sigma}\overline{\underset{\theta}{\tau}} = \underset{\theta}{\tau} + n(n-1)\omega.$$

Consequently, we obtain that

$$\overline{\underset{\theta}{Y}}(x,y)z = \underset{\theta}{Y}(x,y)z, \quad \theta\{1,\ldots,5\}, \tag{16}$$

where the tensor $\underset{\theta}{Y}(x,y)z$ is defined by (14) in the space $(M,g)$ and the tensor $\overline{\underset{\theta}{Y}}(x,y)z$ is defined in the same manner in the space $(\overline{M},\overline{g})$.

From (16) it follows that

$$\overline{\underset{\theta}{E}}(x,y) = \underset{\theta}{E}(x,y), \quad \theta\{1,\ldots,5\},$$

where the tensor $\underset{\theta}{E}(x,y)$ is defined by (15) in the space $(M,g)$ and the tensor $\overline{\underset{\theta}{E}}(x,y)$ is defined in the same manner in the space $(\overline{M},\overline{g})$. $\square$

The tensors $\underset{\theta}{Y}(x,y)z$ and $\underset{\theta}{E}(x,y)z$, $\theta = 1,\ldots,5$ determined in Theorem 4 by (14) and (15) are analogous to Yano's tensor of concircular curvature and the Einstein tensor [11], respectively.

**Theorem 5.** *If there exists $\theta \in \{1,\ldots,5\}$ such that the scalar curvature $\underset{\theta}{\tau}$ of a generalized Riemannian space $(M,g)$ is constant, then the tensor $\underset{\theta}{Y}(x,y)z$ determined by (14) satisfies*

$$(\underset{\eta}{\nabla}_v \underset{\theta}{Y})(x,y)z = (\underset{\eta}{\nabla}_v \underset{\theta}{R})(x,y)z, \quad \eta \in \{0,\ldots,4\},$$

*where $x,y,z,v \in T_p(M)$.*

**Remark 1.** *The result given in Theorem 5 is a generalization of the well-known result which is valid in a Riemannian space:*

$$(\underset{0}{\nabla}_v Y)(x,y)z = (\underset{0}{\nabla}_v R)(x,y)z,$$

*where $Y(x,y)z$ is a tensor of concircular curvature and $R(x,y)z$ is a Riemannian tensor of a Riemannian space.*

## 3. Eisenhart Symmetric Spaces

In this section we shall consider a generalization of symmetric spaces in the settings of generalized Riemannian spaces in Eisenhart's sense.

**Definition 2.** *An n-dimensional $(n > 2)$ generalized Riemannian space $(M,g)$ is said to be an hlEisenhart symmetric space of type $(\eta,\theta) \in \{0,\ldots,4\} \times \{1,\ldots,5\}$ if the curvature tensor $\underset{\theta}{R}(x,y)z$ satisfies*

$$(\underset{\eta}{\nabla}_v \underset{\theta}{R})(x,y)z = 0,$$

*where $x,y,z,v \in T_p(M)$.*

In the following theorems we will derive expressions of the covariant derivatives of tensors that are analogous to Yano's tensor of concircular curvature in various Eisenhart symmetric spaces.

**Theorem 6.** *Let $(M,g)$ be an n-dimensional $(n>2)$ Eisenhart symmetric space of type $(0,\theta)$, $\theta \in \{1,\dots,5\}$. If the scalar curvature $\underset{\theta}{\tau}$ of the space $(M,g)$ is constant, then the covariant derivatives $\underset{\eta}{\nabla}_l \underset{\theta}{Y}^h_{ijk}$, $\eta = 0,\dots,4$ are described as*

$$\underset{0}{\nabla}_l \underset{\theta}{Y}^h_{ijk} = 0,$$

$$\underset{1}{\nabla}_l \underset{\theta}{Y}^h_{ijk} = \partial_l \underset{\theta}{R}^h_{ijk} + \frac{1}{2}T^h_{pl}\underset{\theta}{R}^p_{ijk} - \frac{1}{2}T^p_{il}\underset{\theta}{R}^h_{pjk} - \frac{1}{2}T^p_{jl}\underset{\theta}{R}^h_{ipk} - \frac{1}{2}T^p_{kl}\underset{\theta}{R}^h_{ijp},$$

$$\underset{2}{\nabla}_l \underset{\theta}{Y}^h_{ijk} = \partial_l \underset{\theta}{R}^h_{ijk} + \frac{1}{2}T^h_{lp}\underset{\theta}{R}^p_{ijk} - \frac{1}{2}T^p_{li}\underset{\theta}{R}^h_{pjk} - \frac{1}{2}T^p_{lj}\underset{\theta}{R}^h_{ipk} - \frac{1}{2}T^p_{lk}\underset{\theta}{R}^h_{ijp},$$

$$\underset{3}{\nabla}_l \underset{\theta}{Y}^h_{ijk} = \partial_l \underset{\theta}{R}^h_{ijk} + \frac{1}{2}T^h_{pl}\underset{\theta}{R}^p_{ijk} - \frac{1}{2}T^p_{li}\underset{\theta}{R}^h_{pjk} - \frac{1}{2}T^p_{lj}\underset{\theta}{R}^h_{ipk} - \frac{1}{2}T^p_{lk}\underset{\theta}{R}^h_{ijp},$$

$$\underset{4}{\nabla}_l \underset{\theta}{Y}^h_{ijk} = \partial_l \underset{\theta}{R}^h_{ijk} + \frac{1}{2}T^h_{lp}\underset{\theta}{R}^p_{ijk} - \frac{1}{2}T^p_{il}\underset{\theta}{R}^h_{pjk} - \frac{1}{2}T^p_{jl}\underset{\theta}{R}^h_{ipk} - \frac{1}{2}T^p_{kl}\underset{\theta}{R}^h_{ijp}.$$

**Proof.** Let us outline the proof of the first relation. From Theorem 5 we have that

$$(\underset{1}{\nabla}_v \underset{\theta}{Y})(x,y)z = (\underset{1}{\nabla}_v \underset{\theta}{R})(x,y)z,$$

which together with (1) and $(\underset{0}{\nabla}_v R)(x,y)z = 0$ provides the proof of the first part of this theorem. The rest of the proof can be completed analogously. □

In the same manner we can prove the following theorems.

**Theorem 7.** *Let $(M,g)$ be an n-dimensional $(n>2)$ Eisenhart symmetric space of type $(1,\theta)$, $\theta \in \{1,\dots,5\}$. If the scalar curvature $\underset{\theta}{\tau}$ of the space $(M,g)$ is constant, then the covariant derivatives $\underset{\eta}{\nabla}_l \underset{\theta}{Y}^h_{ijk}$, $\eta = 0,\dots,4$ are described as*

$$\underset{0}{\nabla}_l \underset{\theta}{Y}^h_{ijk} = \partial_l \underset{\theta}{R}^h_{ijk} + \frac{1}{2}T^h_{lp}\underset{\theta}{R}^p_{ijk} - \frac{1}{2}T^p_{li}\underset{\theta}{R}^h_{pjk} - \frac{1}{2}T^p_{lj}\underset{\theta}{R}^h_{ipk} - \frac{1}{2}T^p_{lk}\underset{\theta}{R}^h_{ijp},$$

$$\underset{1}{\nabla}_l \underset{\theta}{Y}^h_{ijk} = 0,$$

$$\underset{2}{\nabla}_l \underset{\theta}{Y}^h_{ijk} = \partial_l \underset{\theta}{R}^h_{ijk} + T^h_{lp}\underset{\theta}{R}^p_{ijk} - T^p_{li}\underset{\theta}{R}^h_{pjk} - T^p_{lj}\underset{\theta}{R}^h_{ipk} - T^p_{lk}\underset{\theta}{R}^h_{ijp},$$

$$\underset{3}{\nabla}_l \underset{\theta}{Y}^h_{ijk} = \partial_l \underset{\theta}{R}^h_{ijk} - T^p_{li}\underset{\theta}{R}^h_{pjk} - T^p_{lj}\underset{\theta}{R}^h_{ipk} - T^p_{lk}\underset{\theta}{R}^h_{ijp},$$

$$\underset{4}{\nabla}_l \underset{\theta}{Y}^h_{ijk} = \partial_l \underset{\theta}{R}^h_{ijk} + T^h_{lp}\underset{\theta}{R}^p_{ijk}.$$

**Theorem 8.** *Let $(M,g)$ be an n-dimensional $(n>2)$ Eisenhart symmetric space of type $(2,\theta)$, $\theta \in \{1,\dots,5\}$. If the scalar curvature $\underset{\theta}{\tau}$ of the space $(M,g)$ is constant, then the covariant derivatives $\underset{\eta}{\nabla}_l \underset{\theta}{Y}^h_{ijk}$, $\eta = 0,\dots,4$ are described as*

$$\underset{0}{\nabla}_l \underset{\theta}{Y}^h_{ijk} = \partial_l \underset{\theta}{R}^h_{ijk} + \frac{1}{2}T^h_{pl}\underset{\theta}{R}^p_{ijk} - \frac{1}{2}T^p_{il}\underset{\theta}{R}^h_{pjk} - \frac{1}{2}T^p_{jl}\underset{\theta}{R}^h_{ipk} - \frac{1}{2}T^p_{kl}\underset{\theta}{R}^h_{ijp},$$

$$\underset{1}{\nabla}_l \underset{\theta}{Y}^h_{ijk} = \partial_l \underset{\theta}{R}^h_{ijk} + T^h_{pl}\underset{\theta}{R}^p_{ijk} - T^p_{il}\underset{\theta}{R}^h_{pjk} - T^p_{jl}\underset{\theta}{R}^h_{ipk} - T^p_{kl}\underset{\theta}{R}^h_{ijp},$$

$$\underset{2}{\nabla}_l \underset{\theta}{Y}^h_{ijk} = 0,$$

$$\underset{3}{\nabla}_l \underset{\theta}{Y}^h_{ijk} = \partial_l \underset{\theta}{R}^h_{ijk} + T^h_{pl}\underset{\theta}{R}^p_{ijk},$$

$$\underset{4}{\nabla}_l \underset{\theta}{Y}^h_{ijk} = \partial_l \underset{\theta}{R}^h_{ijk} - T^p_{il}\underset{\theta}{R}^h_{pjk} - T^p_{jl}\underset{\theta}{R}^h_{ipk} - T^p_{kl}\underset{\theta}{R}^h_{ijp}.$$

**Theorem 9.** *Let $(M, g)$ be an $n$-dimensional $(n > 2)$ Eisenhart symmetric space of type $(3, \theta)$, $\theta \in \{1, \ldots, 5\}$. If the scalar curvature $\underset{\theta}{\tau}$ of the space $(M, g)$ is constant, then the covariant derivatives $\underset{\eta}{\nabla}_l \underset{\theta}{Y}^h_{ijk}$, $\eta = 0, \ldots, 4$ are described as*

$$\underset{0}{\nabla}_l \underset{\theta}{Y}^h_{ijk} = \partial_l \underset{\theta}{R}^h_{ijk} + \frac{1}{2} T^h_{lp} \underset{\theta}{R}^p_{ijk} - \frac{1}{2} T^p_{il} \underset{\theta}{R}^h_{pjk} - \frac{1}{2} T^p_{jl} \underset{\theta}{R}^h_{ipk} - \frac{1}{2} T^p_{kl} \underset{\theta}{R}^h_{ijp},$$

$$\underset{1}{\nabla}_l \underset{\theta}{Y}^h_{ijk} = \partial_l \underset{\theta}{R}^h_{ijk} - T^p_{il} \underset{\theta}{R}^h_{pjk} - T^p_{jl} \underset{\theta}{R}^h_{ipk} - T^p_{kl} \underset{\theta}{R}^h_{ijp},$$

$$\underset{2}{\nabla}_l \underset{\theta}{Y}^h_{ijk} = \partial_l \underset{\theta}{R}^h_{ijk} + T^h_{lp} \underset{\theta}{R}^p_{ijk},$$

$$\underset{3}{\nabla}_l \underset{\theta}{Y}^h_{ijk} = 0,$$

$$\underset{4}{\nabla}_l \underset{\theta}{Y}^h_{ijk} = \partial_l \underset{\theta}{R}^h_{ijk} + T^h_{lp} \underset{\theta}{R}^p_{ijk} - T^p_{il} \underset{\theta}{R}^h_{pjk} - T^p_{jl} \underset{\theta}{R}^h_{ipk} - T^p_{kl} \underset{\theta}{R}^h_{ijp}.$$

**Theorem 10.** *Let $(M, g)$ be an $n$-dimensional $(n > 2)$ Eisenhart symmetric space of type $(4, \theta)$, $\theta \in \{1, \ldots, 5\}$. If the scalar curvature $\underset{\theta}{\tau}$ of the space $(M, g)$ is constant, then the covariant derivatives $\underset{\eta}{\nabla}_l \underset{\theta}{Y}^h_{ijk}$, $\eta = 0, \ldots, 4$ are described as*

$$\underset{0}{\nabla}_l \underset{\theta}{Y}^h_{ijk} = \partial_l \underset{\theta}{R}^h_{ijk} + \frac{1}{2} T^h_{pl} \underset{\theta}{R}^p_{ijk} - \frac{1}{2} T^p_{li} \underset{\theta}{R}^h_{pjk} - \frac{1}{2} T^p_{lj} \underset{\theta}{R}^h_{ipk} - \frac{1}{2} T^p_{lk} \underset{\theta}{R}^h_{ijp},$$

$$\underset{1}{\nabla}_l \underset{\theta}{Y}^h_{ijk} = \partial_l \underset{\theta}{R}^h_{ijk} + T^h_{pl} \underset{\theta}{R}^p_{ijk},$$

$$\underset{2}{\nabla}_l \underset{\theta}{Y}^h_{ijk} = \partial_l \underset{\theta}{R}^h_{ijk} - T^p_{li} \underset{\theta}{R}^h_{pjk} - T^p_{lj} \underset{\theta}{R}^h_{ipk} - T^p_{lk} \underset{\theta}{R}^h_{ijp},$$

$$\underset{3}{\nabla}_l \underset{\theta}{Y}^h_{ijk} = \partial_l \underset{\theta}{R}^h_{ijk} + T^h_{pl} \underset{\theta}{R}^p_{ijk} - T^p_{li} \underset{\theta}{R}^h_{pjk} - T^p_{lj} \underset{\theta}{R}^h_{ipk} - T^p_{lk} \underset{\theta}{R}^h_{ijp},$$

$$\underset{4}{\nabla}_l \underset{\theta}{Y}^h_{ijk} = 0.$$

## 4. Conclusions

In this article we studied conformal and concircular mappings between generalized Riemannian spaces as well as Eisenhart symmetric spaces of various kinds. We started with five linearly independent curvature tensors and obtained the tensors that are analogus to the Weyl conformal curvature tensor, Yano's tensor of concircular curvature and the Einstein tensor. Further, we studied some properties of these tensors as well as the covariant derivatives of tensors analogous to Yano's tensor of concircular in Eisenhart symmetric spaces.

**Author Contributions:** All authors have equally contributed to this work. All authors wrote, read, and approved the final manuscript.

**Funding:** The research by Miloš Z. Petrović and Mića S. Stanković leading to these results was partially supported by the Ministry of Education, Science and Technological Development of the Republic of Serbia, project No. 174012 and the research by Patrik Peška leading to these results has received funding from IGA_PrF_2019_015 Palacký University, Olomouc, Czech Republic.

**Acknowledgments:** The authors would like to thank the referees for their valuable comments which helped to improve the manuscript.

**Conflicts of Interest:** The authors declare no conflict of interest.

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
