# Peer review of "On Conformal and Concircular Diffeomorphisms of Eisenhart’s Generalized Riemannian Spaces"

_mathematics, doi:10.3390/math7070626_

Round 1

Reviewer 1 Report

The main question addressed by research was to describe tensors invariant with respect to conformal and concircular mappings between Eisenhart's generalized Riemannian spaces.

It is relevant and interesting, because the topic of studying mappings between Riemannian spaces and their generalizations is active and has applications.

The topic is originsal and develops important ideas from Riemannian geometry and mathematical physics as well.

The paper adds a new approach in studying diffeomorphisms between generalized Riemannian spaces. Also, it introduces Eisenhart symmetric generalized Riemannian spaces as new symmetric spaces with torsion.

The conclusions are consistent with the evidence as well as with given arguments and address the main question and open new direction of studying mappings between Riemannian spaces without additional conditions.

By my opinion, the paper is written in clear and concise way and contains interesting results. I have no other comments for authors.

Author Response

The main question addressed by research was to describe tensors

invariant with respect to conformal and concircular mappings between Eisenhart’s

generalized Riemannian spaces.

It is relevant and interesting, because the topic of studying mappings between

Riemannian spaces and their generalizations is active and has applications.

The topic is originsal and develops important ideas from Riemannian geometry

and mathematical physics as well. The paper adds a new approach in studying

diffeomorphisms between generalized Riemannian spaces. Also, it introduces

Eisenhart symmetric generalized Riemannian spaces as new symmetric spaces with

torsion.

The conclusions are consistent with the evidence as well as with given arguments

and address the main question and open new direction of studying mappings between

Riemannian spaces without additional conditions. By my opinion, the paper

is written in clear and concise way and contains interesting results. I have no other

comments for authors.

Reviewer 2 Report

This paper studies certain geometrical aspects of connections and conformal relatedness. I think it has relatively low interest. It could, perhaps, be improved and made publishable but only if the authors attend to the following points.

The mapping between manifolds which they start with surely must be a bijection and this should be stated at the beginning. 

Their definition of a conformal mapping is confusing. They start with two manifolds and a map f between them and set up coordinates "pointwise" under f. Then they say that they can continue with just one manifold M. Okay. But then they say that if metrics g and g' on M are conformally related this makes f a conformal mapping. Yes, in a sense , this is correct. But the conformal nature of f is really got from the pullback of g' under f. It is an unusual way to do things! Later in the paper they go back to two manifolds!!

On page 2 are the metrics g(under bar) and g (under v) non-degenerate?----in fact, are they metrics? Do many people work on non-symmetric metrics these days?

At the bottom of page 2, I assume the authors mean that the skew parts of the metrics may not be conformally related (if they are metrics). Maybe they should clarify this.

In definition 2.1 they suddenly introduce five types of Ricci tensor. I assume this comes from the 5 different covariant derivatives they mention later. But this is not well known and should, at least, be explained. However, what is the geometrical significance of these various covariant derivatives? Are they of sufficient importance to be studied to this extent? 

On page 3 (bottom) isn't this essentially the Weyl conformal tensor?  Maybe they should say this clearly. And later (page 5) are they not studying the algebraic symmetries of the Weyl/Riemann tensors?

The tensors Y and E introduced later seem to look like the (curvature) expressions for the (deviation from) constant curvature and (deviation from) the Einstein space condition.

In remark (2.1) I think they mean THE result..........

In this paper the authors talk about the concircular condition. Strangely they give two definitions of it but do not relate them!  In 2.2 are they not merely saying that d(\sigma) is a gradient?

In view of the relatively low interest in the multitude of covariant derivatives in this paper I think the authors should either tell us what these covariant derivatives mean or considerably shorten the paper in order to (only) state their results----nothing more. Whilst I agree it is important to know the various tensors which are unchanged under certain other changes (e.g. the invariance of the Weyl conformal tensor under conformal metric changes) I think that maybe they are taking this a little too far. I encourage them to reconsider the paper and maybe resubmit it when it is rewritten.

Author Response

We rewrite the manuscript by considering all remarks and suggestions

given by the reviewer.

Round 2

Reviewer 2 Report

I think the authors have made an effort to correct the paper and I have no further comments on it.